# Responses of Manila Grass (*Zoysia matrella*) to chilling stress: From transcriptomics to physiology

**Sixin Long[1], Fengying Yan[1], Lin Yang[1], Zhenyuan Sun[2]\*, Shanjun Wei[1]\***

1 College of Life & Environmental Science, Minzu University of China, Beijing, PR China, 2 Research Institute of Forestry, Chinese Academy of Forestry, Beijing, China

\* lkyszy@126.com (ZS); Wei.s.j@163.com (SW)

**Data Availability Statement:** All sequences were deposited in the Short Read Achieve (SRA) division of the Genbank repository (accession no. SRS1819789).

## Abstract

Manila grass (*Zoysia matrella*), a warm-season turfgrass, usually wilts and browns by late autumn because of low temperature. To elucidate the molecular mechanisms regarding Manila grass responses to cold stress, we performed transcriptome sequencing of leaves exposed to 4°C for 0 (CK), 2h (2h_CT) and 72h (72h_CT) by Illumina technology. Approximately 250 million paired-end reads were obtained and de novo assembled into 82,605 unigenes. A total of 34,879 unigenes were annotated by comparing their sequence to public protein databases. At the 2h- and 72h-cold time points, 324 and 5,851 differentially expressed genes (DEGs) were identified, respectively. Gene ontology (GO) and metabolism pathway (KEGG) enrichment analyses of DEGs indicated that auxin, gibberellins, ethylene and calcium took part in the cold signal transduction in the early period. And in the late cold period, electron transport activities, photosynthetic machinery and activity, carbohydrate and nitrogen metabolism, redox equilibrium and hormone metabolism were disturbed. Low temperature stress triggered high light, drought and oxidative stress. At the physiological level, cold stress induced a decrease in water content, an increase in levels of total soluble sugar, free proline and MDA, and changes in bioactive gibberellins levels, which supported the changes in gene expression. The results provided a large set of sequence data of Manila grass as well as molecular mechanisms of the grass in response to cold stress. This information will be helpful for future study of molecular breeding and turf management.

## Introduction

Low temperature is one of primary abiotic stresses limiting the growth and geographical distribution of plants. Sophisticated biochemical and physiological modifications occur in plants undergoing low temperature stimulus, such as increases in abscisic acid, soluble sugars and free proline, alterations in photosynthesis, protein translation and respiration. Changes in energy conservation and metabolic homeostasis contribute to poor germination, stunted seedlings, chlorosis, wilting, and reduced leaf expansion, and may also lead to tissue death

**Funding:** This work was funded by the National Natural Science Foundation of China (31100507) to SW and the Fundamental Research Funds for the Central University.

**Competing interests:** The authors have declared that no competing interests exist.

(necrosis) [1]. Plant species and cultivars are diverse greatly in cold response. Generally, plants inhabiting temperate regions, such as Arabidopsis and wheat, are tolerant to chilling tolerance and can acquire freezing tolerance during exposure to chilling and non-freezing temperatures. The process is called cold acclimation. By contrast, plants adapting to tropical and subtropical origins, such as maize and foxtail millet, are sensitive to chilling temperatures (0–10˚C) and are unable to cold acclimate. For cold sensitive species, however, chilling tolerance can also be enhanced by pre-exposure to a higher chilling temperature for a certain period of time [2].

There is a clear association between adaptability of an organism and plasticity of gene expression. In past decades, the molecular mechanism of cold or chilling acclimation and acquired cold tolerance in plants, mainly in Arabidopsis (*Arabidopsis thaliana*) and some crop species, including wheat (*Triticum aestivum*), rice (*Oryza sativa*), maize (*Zea mays*), and tomato *(Lycopersicon esculentum)*, has been extensively studied. For instance, approximately 1000 genes in Arabidopsis [3, 4], more than 2% of the genome of wheat [5], and more than 2500 genes in maize [2], have been found to be cold regulated. These genes can be classified into two groups according to biological function [4, 6–7]. The first group includes genes of osmolyte biosynthesis, antioxidant enzymes and dehydrins that protect cells from stresses directly. The second group consists of genes that transduce signal and regulate transcription, such as protein kinases and transcription factors (TFs). The ICE-CBF-COR is currently the best documented pathway involved in cold response. ICE (**i**nducer of **C**BF **e**xpression) is activated by phosphorylation under cold stress and then promote the expression of CBFs (**C**-repeat (CRT)-**b**inding **f**actors) rapidly. The CBF proteins recognize the promoter regions of multiple cold-regulated (COR) genes to activate their transcription [4]. Transcript profiling revealed that a percent of 11% COR genes are target of CBFs and extensively function in the transcription regulation, carbohydrate and lipid metabolism and cell wall modification [6, 8]. Based on genome wide analyses, more TFs have been found to be involved in the cold response, such as WRKY [9], NAC [10], MYB [11] and AP2/ERF [12]. Furthermore, the co-regulation of gene expression by multiple TFs is extensive, which is supported by the evidence that about 25% of CBF2-targeted genes are also regulated by five first-wave transcription factors in cold, namely HSFC1, ZF, CZF1, ZAT10 and ZAT12 [6, 13].

With the development of sequencing techniques, the mechanisms of cold response in plants without a reference genome, in addition to model plants, have been investigated in depth. RNA sequencing has been applied successfully in many plants such as lily (*Lilium lancifolium*) [14], chrysanthemum (*Chrysanthemum morifolium*) [15], tea (*Camellia sinensis*) [16], perennial ryegrass (*Lolium perenne* L.) [17] and bermudagrass (*Cynodon dactylon*) [18], to detect and elucidate the breadth of molecular mechanisms involved in cold response. Some specific cold adjustments were revealed at the transcriptome level. For instance, karrikins, a new group of substances existing in the smoke of burning plant materials, have been found to be involved in the response to cold stress [19]. Further, in perennial ryegrass the transcriptional mechanisms during cold acclimation that underlie carbon allocation in fructan biosynthesis, an important response in abiotic stress tolerance in temperate grasses, have been described [17]. The transcriptome data from multiple plant species will facilitate our understanding of the cold response in a wider range of plants.

Turgrfasses has significant ecological, environmental, and economic impacts since they provide a ground cover in home lawn, sports field, and landscapes. Low temperature stress always limits the usage of turfgrass species. In recent decades, significant progress has been achieved in turfgrass stress physiology and molecular biology, but research for turfgrasses at the molecular and genomic levels generally lags far behind that of the major Poaceae [20]. For instance, Manila grass (*Zoysia matrella* Merr.), a popular warm season perennial turfgrass, shows excellent tolerance to drought and high temperature stresses, but is sensitive to low

temperature. The turf it builds always exhibits a long yellow period in temperate zones or even dies off during winter freezes, which compromises its ornamental value and usage greatly. Improving cold tolerance and prolonging the green period is important for maintaining the use of the grass where it is currently grown, and for expanding it into northern areas. A comprehensive understanding of the cold response characteristics at the molecular level will be of great help in this regard. Previous studies have elucidated some physiological and metabolic changes in Manila grass exposed to low temperature, including proline and water-soluble carbohydrate levels and cell leakage [21, 22]. At the protein level, Xuan (2013) reported that some proteins were differentially accumulated in stolons between *Z.japonica* and *Z.metrella* after a 14d-cold exposure [23]. The gene expression profiles and the molecular mechanism for physiological changes under low temperature conditions are largely unknown.

In the present study, cDNA libraries obtained from leaves of Manila grass grown at normal growth temperature or treated with a 4˚C-cold stress were sequenced using Illumina NGS technology. Since a dynamic series of changes in the plant transcriptome is set in motion upon transfer of plants from warm to cold temperature, in Arabidopsis transient and sustained responses were classified into three major categories: early (occurring over 1–4 h), intermediate (occurring over 12–24 h) and late (occurring over 48–96 h) [3, 6]. According to this, we set 2h- and 72h-cold time points, to explore early and late response respectively, in this study. All sequences were deposited in the Short Read Achieve (SRA) division of the Genbank repository (accession no. SRS1819789). Classical physiological indicators involved in the stress response, including water content, levels of free proline, total soluble sugar and malondialdehyde (MDA) were measured. The content of the cold related phytohormone gibberellin (GA) was also determined. The results provide a large amount of EST resources for Manila grass as well as information about gene expression changes in response to cold stress. GO and KEGG analysis of the differentially expressed genes provides a comprehensive understanding of the cold response of this grass.

## Materials and methods

### Plant material and treatment

Manila grass (*Zoysia matrella*) was planted in plastic pots (10 cm diameter and 15 cm tall) filled with brown coal soil: perlite: roseite (2:1:1). For cold treatment, plants were transferred to a growth chamber set to a 14h-photoperiod, photosynthetically active radiation at 450 μmol m$^{-2}$ s$^{-1}$ at the canopy level, and a temperature of 4/6˚C (day/night) for 2h (2h_CT) and 72h (72h_CT). Plants grown at 25/20˚C (day/night) were used as the control. The top two expanded leaves from three pots (three replicates) were frozen immediately in liquid nitrogen, and then stored at −80˚C prior to RNA-Seq analysis.

### Total RNA extraction, RNA-seq library construction and sequencing

Total RNA was extracted from leaves with the Universal Plant Total RNA Extraction Kit RP3301 (BioTake Corporation, Beijing, China) according to the manufacturer's instructions. RNA samples with a RIN value above 7, a 28S:18S ratio higher than 1.5:1 and total amount exceeding 0.1μg were acceptable. A cDNA library was constructed as described earlier [24] and then sequenced. Two experiments were conducted independently. The sequencing work of the first experiment was conducted by Capitalbio Company (Beijing, China) with a HiSeq2000 system, one repeat for each treatment. The work of the second experiment was conducted by Biomarker Technologies Co, LTD (Beijing, China) on the Illumina Hiseq ×10 platform, two repeats per treatment. The raw sequence reads were deposited in the SRA (Short Read Archive) database of NCBI (National Center for Biotechnology Information).

## Reads filtration, assembly and sequence annotation

Before assembly, raw reads were processed by removing adapter sequences in addition to reads containing poly-N or low in quality to produce clean data (clean reads). All clean reads from the nine libraries were merged and assembled to transcripts using Trinity with default parameters (v.r2013-08-14) [25]. Potential alternative spliced isoforms produced during the *de novo* assembly were grouped into a unigene cluster. The longest transcript in a unigene cluster was used to form a set of non-redundant unigene dataset. The function of unigenes were annotated by aligning sequences to the publicly available protein database including Uniprot (Swiss-Prot/TrEMBL), Nr (non-redundant protein sequences in NCBI), KEGG (Kyoto Encyclopedia of Genes and Genomes database), KOG (eukaryotic orthologous groups), COG (clusters of orthologous groups), Pfam (protein families) and Eggnog (evolutionary genealogy of genes: Non-supervised Orthologous Groups), using the BLASTx analysis with the significant threshold of E-value≦1e-5 (v2.2.6). GO (Gene ontology) terms for unigenes were obtained using Blast2GO software [26] which assigned homologous sequences aligned by BLAST with Uniprot and NCBI nr database to GO terms.

## Calculation of gene expression level and identification of cold response genes

Expression quantification and differential expression of genes were analyzed with the Trinity platform. For each sequenced library, transcript abundance was estimated with RSEM (v1.2.6) [27], the expression values were then normalized as FPKM (fragments per kilobase of transcript per million fragments mapped). Differentially expressed genes (DEGs) were identified with edgeR (the negative binomial distribution model with parameters of $|\log_2 (\text{fold change})|$ > 1, p-value < 0.01 and FDR < 0.05) [28]. DEGs identified on both Hiseq ×10 and Hiseq 2000 platforms were considered as final DEGs. GO terms and pathways enriched in the set of differentially expressed genes were calculated by the hypergeometric test [29].

## Quantitative real-time PCR analysis

For first strand cDNA synthesis, 1μg DNase I-treated total RNA was used for reverse transcription (RT) with M-MLV (Promega). A 10-fold dilution of the resulting cDNA was used as a template for quantitative real-time PCR (qRT-PCR). *Actin-1* was selected as an internal control gene, and primers used for qRT-PCR were designed by the Primer 3 website (http://frodo.wi.mit.edu/primer3/). qRT-PCR was performed using the One Step SYBR Prime-Script RT-PCR Kit (TAKARA, Dalian, China) according to the manufacturer's instructions. Products were verified by melting curve analysis. All reactions were performed with at least three biological replicates. The relative expression of the genes was calculated by normalizing the number of target transcript copies to the internal control gene using the comparative ΔΔCt method described earlier [30].

## Physiological detection

Contents of malondialdehyde (MDA) [31], water, total water soluble carbohydrate and free proline [21] were measured using previously described protocols. Quantification of endogenous GAs was performed as described [9]. Three biological replicates were performed for GA determinations, and five for the other measurements. Results were expressed as the mean ± standard error (SE). Statistical analysis was performed by One-Way ANOVA followed by LSD multiple comparison test (p < 0.05) using the SPSS statistical package.

**Table 1. Overview of the RNA-seq reads.**

| Sequencing library | Sequencing platform | Data (bp) | %≥Q30 | Clean reads | mapped reads | mapped rate(%) |
|---|---|---|---|---|---|---|
| CK-1 | HiSeq2000 | 5,152,372,051 | 90.68 | 25,768,383 | 22,280,240 | 86.46 |
| 2h_CT-1 | HiSeq2000 | 5,893,884,079 | 91.52 | 29,476,100 | 25,849,872 | 87.70 |
| 72h_CT-1 | HiSeq2000 | 5,810,287,750 | 93.33 | 29,057,878 | 24,896,786 | 85.68 |
| CK-2 | HiSeq X10 | 5,043,325,898 | 93.68 | 25,216,719 | 19,807,836 | 78.55 |
| CK-3 | HiSeq X10 | 5,363,992,158 | 93.76 | 26,820,055 | 20,967,422 | 78.18 |
| 2h_CT-2 | HiSeq X10 | 6,508,034,994 | 93.82 | 32,540,320 | 25,166,786 | 77.34 |
| 2h_CT -3 | HiSeq X10 | 5,894,858,180 | 93.93 | 29,474,414 | 22,912,252 | 77.74 |
| 72h_CT-2 | HiSeq X10 | 5,480,400,562 | 93.46 | 27,402,090 | 21,179,430 | 77.29 |
| 72h_CT-3 | HiSeq X10 | 5,623,463,266 | 93.68 | 28,117,426 | 21,876,665 | 77.80 |

## Results

### Transcriptome sequencing and reads assembly

An overview of the clean pair-end reads is presented in Table 1. About 25 to 30 million pair-end reads were obtained for each cDNA library. The clean reads from all libraries were subjected to assembly using the Trinity program. A total of 286,574 transcripts were generated (Table 1). The mean transcript size was 1,480 nt, with an N50 of 2,236 nt. All reads were mapped back to the transcripts to assess the efficiency of the assembly. A percent of 77%-87% of reads in all the libraries were mapped back successfully (Table 2). Compared with the reads obtained from the Illumina HiSeq ×10, reads obtained from the Illumina HiSeq 2000 showed a higher mapping rate, indicating the different quality between the two platforms. Potential alternative spliced transcript isoforms were assigned into a unigene cluster and the longest transcripts in each cluster were used to get a non-redundant unigene dataset. In this way, 82,605 unigene clusters were obtained, of which a large part (68.99%) contained only one transcript, while 0.77% contained more than 50 transcripts (Fig 1). For the unigene dataset, the mean length was 787 nt, the N50 was 1,471 nt. A total of 33,507 (40.56%) unigenes were longer than 500 nt, and 8064 (9.76%) unigenes were longer than 2,000 nt (Table 2), suggesting that a batch of transcripts containing complete CDS were obtained.

### Functional annotation and classification of unigenes

We performed BLAST (E-value <1e-5) analysis of all the unigenes against publicly available protein databases: Uniprot, Nr, KEGG, KOG, COG, GO, Pfam and eggNOG. Of all the unigenes, 34,879 (42.22%) significantly matched a sequence in at least one of the above databases. Nr, eggnog and GO were the top three matched databases, to which 40.79%, 38.61% and

**Table 2. Overview of the assembly.**

| Length range(bp) | Transcript | Unigene |
|---|---|---|
| >2000 | 77,273 (26.96%) | 8,064 (9.76%) |
| 1000–2000 | 80,863 (28.22%) | 9,706 (11.75%) |
| 500–1000 | 52,118 (18.19%) | 15,737 (19.05%) |
| 300–500 | 37,710 (13.16%) | 21.121 (25.57%) |
| 200–300 | 38,610(13.47%) | 27,977 (33.87%) |
| Total number | 286,574 | 82,605 |
| N50 | 2,236 | 1,471 |
| Mean length | 1480.03 | 787.86 |

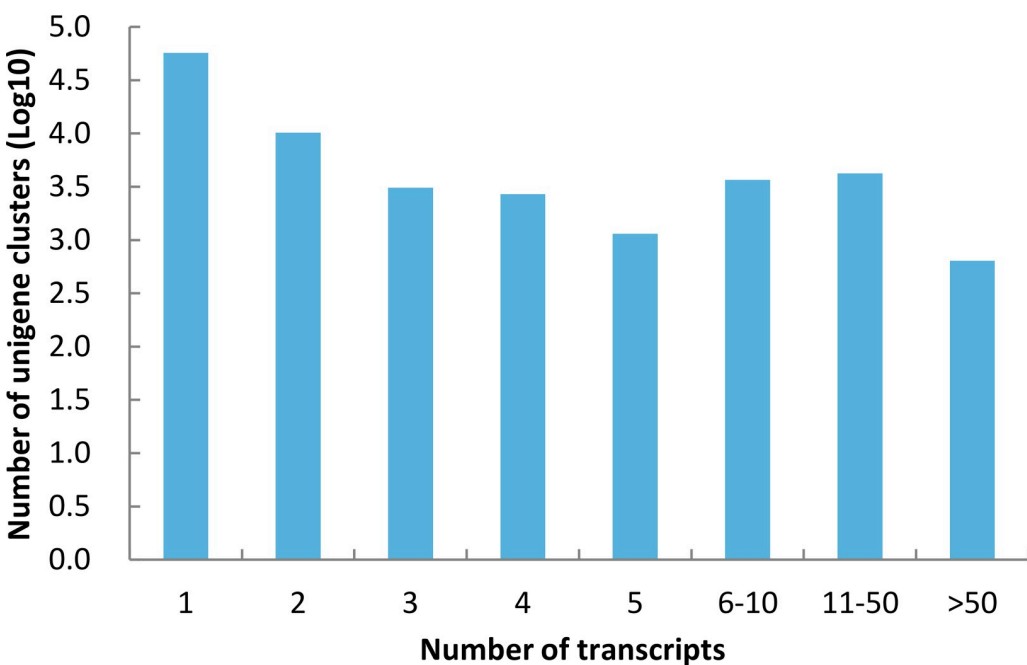

**Fig 1. Histogram of unigene clusters.** The X-axis lists how many transcripts are in a unigene cluster, and the Y-axis lists the number of unigene clusters.

30.91% of unigenes were matched, respectively. Among the annotated unigenes, 36.15% were 300–1000 nt and 43.46% were over 1000 nt in length (Table 3).

## Species distribution of the top BLAST hits

To study the sequence conservation of Manila grass, we used BLAST to align all unigenes to the Uniprot database with an E-value of $1.0 \times 10^{-5}$. A total of 33,476 unigenes had hits with known nucleotide sequences. Of these, 85.0% of them had hits to monocotyledonous species. The closest species was foxtail millet (*Setaria italic*), with 10,769 unigenes (32.17%) matching. The second closest reference species was *Sorghum bicolor*, showing 14.95% homology with Manila grass. *Zea mays* (14.02%), *Oryza* spp. (12.00%), *Triticum aestivum* (3.17%), and *Hordeum vulgare* (2.74%) are reference species in decreasing order (Fig 2). Previous findings also showed that *S. italica* and *S. bicolor* were the top two species homologous to *Zoysia japonica*,

**Table 3. Number of unigenes blasted to databases (E<1.0e-5).**

| Anno_Databae | Annotated _number | 300≦length<1000 | length≥1000 |
|---|---|---|---|
| COG | 10,243 | 2,833 | 5,588 |
| GO | 25,532 | 8,588 | 12,354 |
| KEGG | 11,490 | 3,902 | 5,294 |
| KOG | 17,710 | 5,465 | 8,982 |
| Pfam | 21,933 | 6,539 | 12,266 |
| Swissport | 20,525 | 6,802 | 10,310 |
| eggNOG | 31,890 | 11,160 | 14,831 |
| Nr | 33,697 | 12,097 | 15,068 |
| All annotated | 34,879(42.22%) | 12,613 | 15,161 |

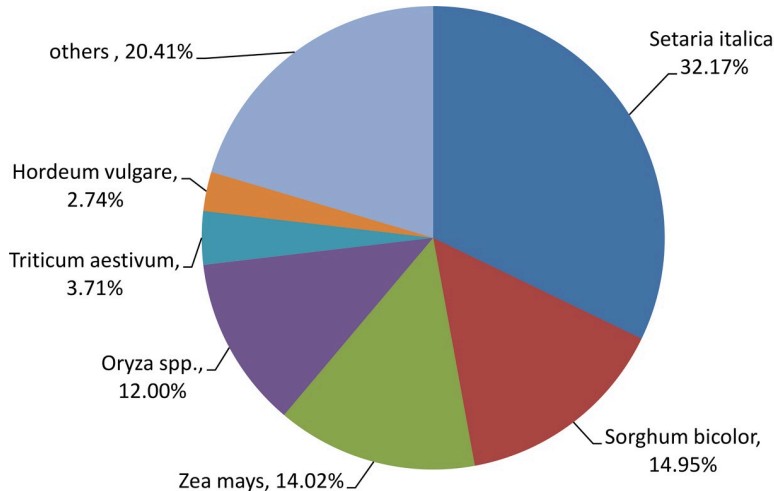

**Fig 2. Similarity of Manila grass sequences with those of other species.**

another species in the *Zoysia* genus [32–34]. These data provide supporting evidence that our transcriptome was precisely assembled.

## Functional classification of unigenes

The potential functions of the Manila grass unigenes were assessed using Blast2GO [35]. A total of 25,532 (30.91%) unigenes were assigned to at least one GO term annotation successfully (Fig 3). The unigenes were then classified into three major GO categories: biological

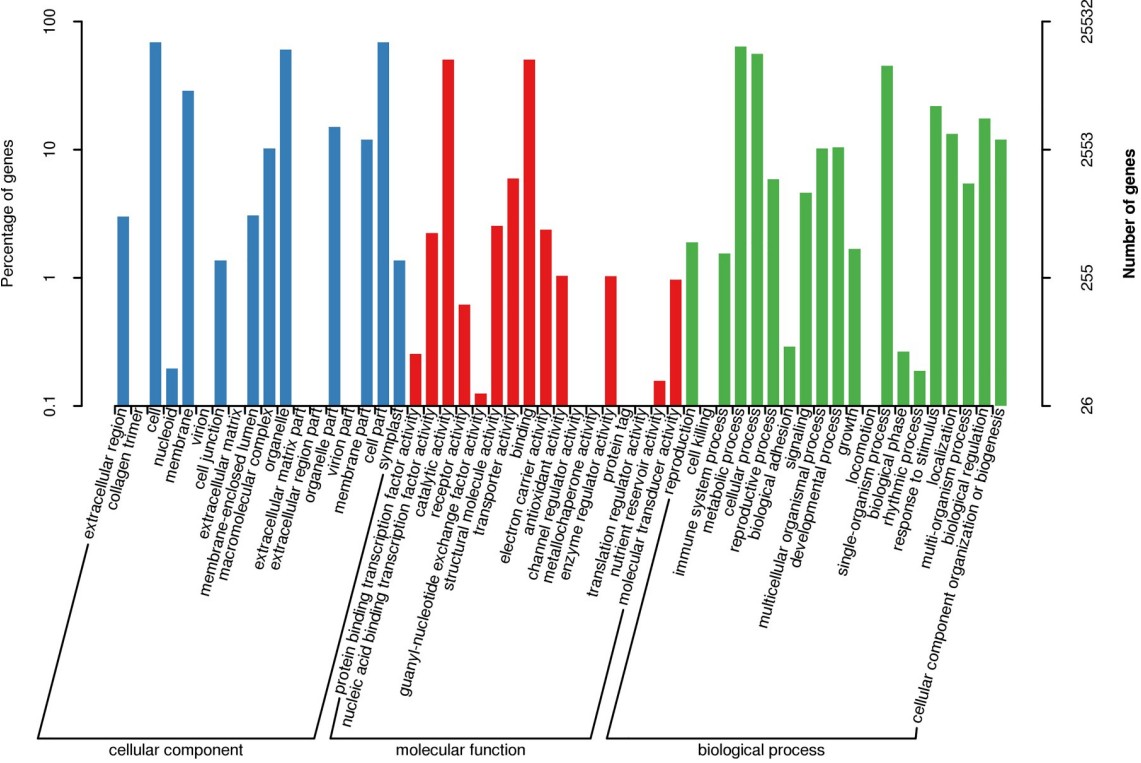

**Fig 3. GO function classification of unigenes.**

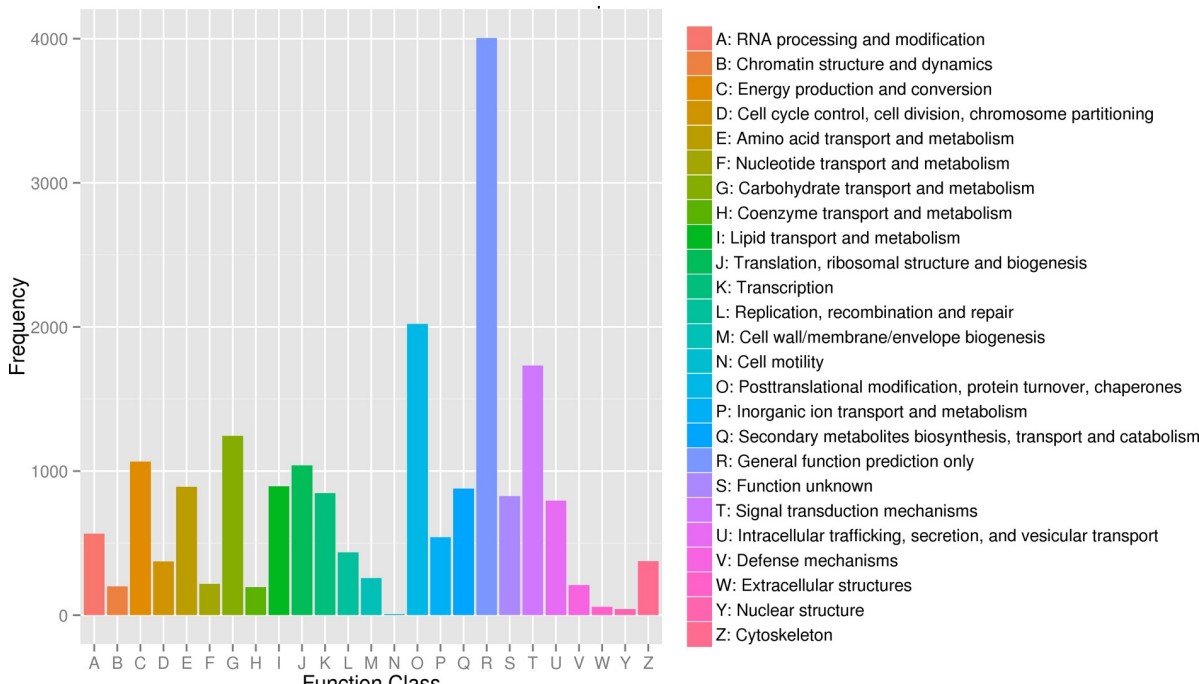

A: RNA processing and modification
B: Chromatin structure and dynamics
C: Energy production and conversion
D: Cell cycle control, cell division, chromosome partitioning
E: Amino acid transport and metabolism
F: Nucleotide transport and metabolism
G: Carbohydrate transport and metabolism
H: Coenzyme transport and metabolism
I: Lipid transport and metabolism
J: Translation, ribosomal structure and biogenesis
K: Transcription
L: Replication, recombination and repair
M: Cell wall/membrane/envelope biogenesis
N: Cell motility
O: Posttranslational modification, protein turnover, chaperones
P: Inorganic ion transport and metabolism
Q: Secondary metabolites biosynthesis, transport and catabolism
R: General function prediction only
S: Function unknown
T: Signal transduction mechanisms
U: Intracellular trafficking, secretion, and vesicular transport
V: Defense mechanisms
W: Extracellular structures
Y: Nuclear structure
Z: Cytoskeleton

**Fig 4. KOG function classification of unigenes.**

processes, cellular components, and molecular function. Among the cellular components category, "cell", "cell part" and "organelle" were the major components, to which 68.97%, 68.97% and 60.41% of the GO-annotated unigenes respectively were assigned. Among the category of molecular functions, the top three components were "catalytic activity", "binding" and "transporter activity", to which 50.54%, 50.33% and 5.97% of the GO-annotated unigenes respectively were assigned. Among the category of biological processes, the top three components were "metabolic process", "cellular process" and "single-organism process", to which 63.75%, 56.07% 45.00% of the GO-annotated unigenes were assigned. The results were in accord with the GO annotation of transcriptome data of *Z. japonica* we reported earlier [33].

A search against the KOG database indicated that 17,710 unigenes had the best hits and were classified into 25 functional categories (Fig 4). "General function prediction only" was the largest group (4,004, 22.61%), followed by "posttranslational modification, protein turnover, chaperone" (2,021, 11.41%), and "signal transduction mechanism" (1,733, 9.79%); the smallest category was "cell motility" (8, 0.05%). The size order of these groups was similar to that of the leaf transcriptome from 'Meyer' zoysiagrass [33], but was different from that of the root transcriptome from 'Zenith' zoysiagrass [34]. 'Meyer' and 'Zenith' are two cultivars of *Zoysia japonica*. It can be concluded that the leaf transcriptomes are highly similar between *Z. japonica* and *Z. metrella*, while the transcriptomes of leaves and roots within a species are different.

## Analysis of differential gene expression

To determine the molecular events induced by low temperature, the RPKM method (Reads Per kb per Million reads) was used to calculate expression levels of the unigenes. Compared with the control, unigenes with FDR < 0.05 and ratios larger than 2 (P<0.01) were considered to be differentially expressed genes (DEG). A total of 6,175 DEGs were identified between

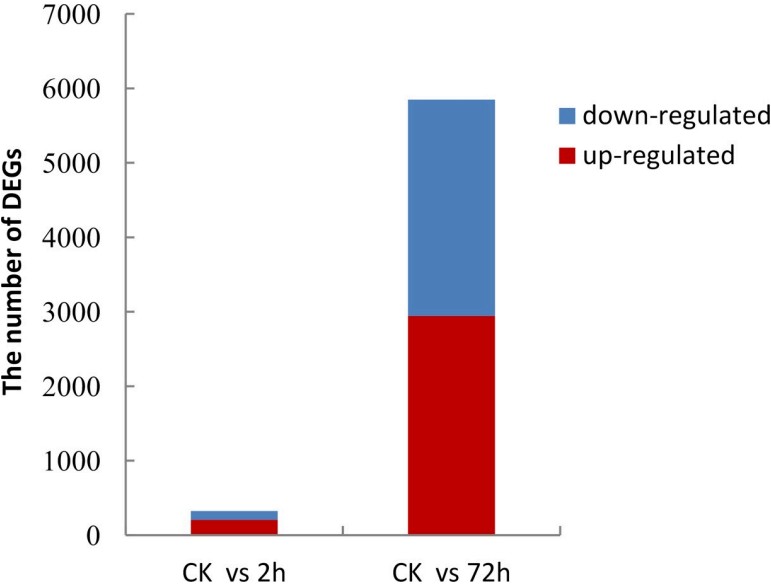

**Fig 5. DEGs in Manila grass leaves under cold stress.**

cold-treated and control libraries. Screening of the DEGs indicated that both up- and down-regulation of gene expression occurred, and that gene expression changed over time during the cold treatment. Among all DEGs, 205 were quickly induced while 119 were down-regulated by cold (Fig 5; S1 Table). After 72h-cold, 2,944 genes were up-regulated, and 2,907 were down-regulated (Fig 6; S2 Table). To validate the expression data obtained from RNA-seq, 15 DEGs were evaluated by qRT-PCR analysis using the primer pairs shown in S3 Table. The PCR products showed the expected fragment size when were detected by agarose gel electrophoresis, indicating the reliability of the assembly. The qRT-PCR results (Table 4) were consistent with the RNA-seq data.

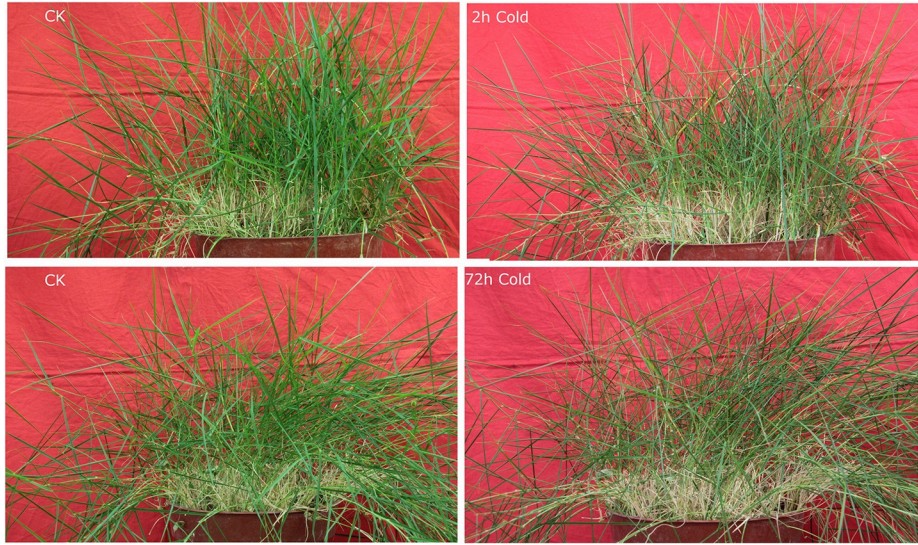

**Fig 6. Morphology of Manila grass under cold stress.**

**Table 4. Verification of RNA-seq results by qRT-PCR.**

| Unigene ID | Annotation | Log FC in RNA-Seq | | | | (-ΔΔ CT) in qRT-PCR | |
|---|---|---|---|---|---|---|---|
| | | 2h_CT Vs CK | | 72h_CT Vs CK | | 2h_CT Vs CK | 72h_CT Vs CK |
| | | HiSeq 2000 | Hiseq X10 | HiSeq 2000 | Hiseq X10 | | |
| c96574.graph_c9 | dehydration-responsive element-binding protein 1A-like | 4.44 | 4.91 | -2.12 | 0.00 | 6.41±0.21 | 1.18±0.20 |
| c85421.graph_c0 | dehydration-responsive element-binding protein 1H-like | 6.19 | 7.28 | NA | NA | 7.52±0.26 | -0.35±0.23 |
| c85421.graph_c1 | dehydration-responsive element-binding protein 1B-like] | 3.93 | 6.84 | -2.18 | NA | 5.83±0.34 | -0.70±0.06 |
| c86189.graph_c0 | Dehydration-responsive element-binding protein 1G | 3.62 | 4.72 | -7.57 | NA | 4.58±0.16 | -0.67±0.29 |
| c86189.graph_c1 | CBF [Zoysia japonica] | 3.34 | 5.71 | -4.80 | NA | 4.02±0.21 | 0.33±0.04 |
| c101456.graph_c0 | late embryogenesis abundant protein, group 3-like isoform X2 | NA* | NA | 12.35 | 7.69 | NA | 6.18±0.41 |
| c74176.graph_c1 | Dehydrin Rab25 | NA | NA | 10.61 | 9.80 | NA | 5.35±0.04 |
| c85261.graph_c0 | dehydrin [Sorghum bicolor] | NA | -0.67 | 11.60 | 6.97 | 1.03±0.06 | 10.61±0.56 |
| c88396.graph_c1 | PREDICTED: dehydrin COR410-like | 1.50 | 0.85 | 11.32 | 8.82 | 1.46±0.16 | 11.67±0.32 |
| c99476.graph_c2 | Ent-copalyl diphosphate synthase 1, chloroplastic (Precursor) | -2.49 | -0.39 | -3.94 | -1.30 | -0.36±0.17 | -2.33±0.31 |
| c92215.graph_c0 | PREDICTED: gibberellin 2-beta-dioxygenase 8-like | NA | NA | 9.99 | 4.34 | NA | 3.18±0.19 |
| c93791.graph_c2 | PREDICTED: gibberellin 2-beta-dioxygenase-like | NA | NA | 6.03 | 3.35 | 0.37±0.16 | 2.58±0.07 |
| c95389.graph_c1 | PREDICTED: gibberellin 2-beta-dioxygenase | -0.06 | -0.80 | 4.98 | 1.60 | 0.21±0.06 | 2.38±0.14 |
| c93791.graph_c0 | gibberellin 2-beta-dioxygenase | -0.77 | 0.86 | 4.49 | 5.59 | 0.10±0.42 | 4.25±0.28 |
| c90155.graph_c0 | glutamine-dependent asparagine synthetase | -1.14 | 0.27 | 4.93 | 7.09 | -0.68±0.37 | 4.07±0.18 |

*NA means data not available

## The functional categorization of DEGs

To categorize the function of DEGs, we annotated them by assigning them gene ontology (GO) terms. Using a hypergeometric test, we performed a GO enrichment analysis to identify significantly enriched GO terms (p-value<0.05) in DEGs. Pathways that displayed significant changes (P-value <0.05) in response to cold stress were identified using the KEGG database.

Among the 324 DEGs between 2h_CT VS CK, 158 were assigned to gene ontology (GO) terms, 47 were assigned to KEGG pathway. Compared with all unigenes that were annotated by GO terms, DEGs enriched 52 terms significantly (S4 Table). Among the type of cellular component, the top two enriched were "cytoplasmic membrane-bounded vesicle" (GO:0016023) and "nucleus" (GO:0005634), followed by "cytoplasm" (GO:0005737) and "extracellular region" (GO:0005576). These data indicated that the modulation in cytoplasmic membrane and nucleus is sensitive to cold, and that extracellular region and cytoplasm were also action sites of the early cold stress. Among the type of biological process, "auxin-activated signaling pathway (GO:0009734)", "abscisic acid catabolic process (GO:0046345)", "gibberellin metabolic process (GO:0009685)" and "ethylene-activated signaling pathway (GO:0009873)" were included; among the type of molecular function, "sequence-specific DNA binding transcription factor activity (GO:0003700)", "calcium ion binding (GO:0005509)" and "calmodulin binding" (GO:0005516) were included. KEGG enrichment analysis indicated that "phenylalanine metabolism" (ko00360), "phenylpropanoid biosynthesis" (ko00940) and "Plant hormone signal transduction (ko04075) were enriched in the DEGs (S5 Table). These results indicated that Manila responded to cold stress after being exposed to 4°C for 2h; auxin, gibberellins, ethylene and calcium took part in the early period of cold signal transduction; and phenylalanine metabolism was hypersensitive to low temperature.

Among the 5,851 DEGs between 72h_CT and CK, 2,810 were assigned to GO terms, 901 were assigned to KEGG pathway. A total of 169 GO terms (S6 Table) and 16 pathways (S7

Table) were enriched in the DEGs. The notable enriched terms and pathway are shown in Table 5. These results indicated that Manila grass exposed to low temperature stress for 72h was simultaneously subjected to drought, high light and oxygen stresses, and that crosstalk

**Table 5. The notable GO terms and KEGG pathways enriched in the DEGs between 72h_CT VS CK.**

| Item type | Item Name | No. DEG | P-value |
|-----------|-----------|---------|---------|
| GO term | *BP: response to cold (GO:0009409); | 76 | 0.0010 |
| | BP:response to stress (GO:0006950); | 68 | 0.0000 |
| | BP:response to high light intensity (GO:0009644); | 33 | 0.0017 |
| | BP:response to water deprivation (GO:0009414); | 42 | 0.0340 |
| | BP:response to heat (GO:0009408); | 43 | 0.0180 |
| | BP:oxidation-reduction process (GO:0055114); | 238 | 0.0426 |
| | BP:response to abscisic acid (GO:0009737); | 51 | 0.0402 |
| | BP:abscisic acid-activated signaling pathway(GO:0009738); | 23 | 0.0055 |
| | BP:response to ethylene (GO:0009723); | 24 | 0.0047 |
| | BP:response to growth hormone (GO:0060416); | 8 | 0.0056 |
| | BP:cytokinin biosynthetic process (GO:0009691); | 9 | 0.0012 |
| | BP:gibberellin catabolic process (GO:0045487); | 4 | 0.0117 |
| | BP:salicylic acid metabolic process (GO:0009696); | 4 | 0.0207 |
| | BP:response to karrikin (GO:0080167); | 41 | 0.0003 |
| | BP:chloroplast organization (GO:0009658); | 24 | 0.0071 |
| | BP:photosynthesis, light harvesting (GO:0009765); | 11 | 0.0063 |
| | BP:photosystem II assembly (GO:0010207); | 30 | 0.0010 |
| | BP:photosynthesis (GO:0015979); | 29 | 0.0061 |
| | BP:fatty acid biosynthetic process (GO:0006633); | 29 | 0.0345 |
| | BP:nitrogen fixation (GO:0009399); | 4 | 0.0329 |
| | BP:L-asparagine biosynthetic process (GO:0070981); | 3 | 0.0462 |
| | BP:L-serine biosynthetic process (GO:0006564); | 4 | 0.0057 |
| | BP:response to fructose (GO:0009750); | 23 | 0.0009 |
| | BP:response to glucose (GO:0009749); | 19 | 0.0047 |
| | BP:response to sucrose (GO:0009744); | 31 | 0.0198 |
| | BP:transmembrane transport (GO:0055085); | 108 | 0.0010 |
| | BP:carbohydrate transport (GO:0008643); | 22 | 0.0004 |
| | BP:nitrate transport (GO:0015706); | 13 | 0.0377 |
| | BP:proline transport (GO:0015824); | 10 | 0.0004 |
| | BP:oligopeptide transport (GO:0006857); | 19 | 0.0008 |
| | CC:integral component of membrane (GO:0016021); | 418 | 0.0000 |
| | CC:chloroplast thylakoid membrane (GO:0009535); | 76 | 0.0005 |
| | CC:chloroplast thylakoid lumen (GO:0009543); | 19 | 0.0009 |
| | CC:photosystem II oxygen evolving complex GO:0009654); | 12 | 0.0111 |
| | CC:itochondrion (GO:0005739); | 610 | 0.0184 |
| | MF:sequence-specific DNA binding transcription factor activity (GO:0003700); | 112 | 0.0003 |
| | MF:protein disulfide oxidoreductase activity (GO:0015035); | 23 | 0.0066 |
| | MF:oxidoreductase activity (GO:0016491); | 70 | 0.0317 |
| | MF:hydrolase activity (GO:0016787); | 67 | 0.0400 |
| | MF:transferase activity, transferring glycosyl groups (GO:0016757); | 35 | 0.0496 |

(*Continued*)

**Table 5.** (Continued)

| Item type | Item Name | No. DEG | P-value |
|---|---|---|---|
| pathway | Plant hormone signal transduction(ko04075) | 59 | 0.0016 |
| | Porphyrin and chlorophyll metabolism(ko00860) | 17 | 0.0023 |
| | Photosynthesis—antenna proteins(ko00196) | 9 | 0.0271 |
| | Photosynthesis(ko00195) | 21 | 0.0025 |
| | Phenylalanine, tyrosine and tryptophan biosynthesis(ko00400) | 18 | 0.0174 |
| | Glutathione metabolism(ko00480) | 30 | 0.0482 |

existed between cold and heat stress. Plant growth regulators including cytokinin, gibberellin, ethylene, abscisic acid, and salicylic acid took part in the cold signal transduction pathway. Cold stress caused large changes in the expression of genes involved in the photosynthetic apparatus and activity, in nitrogen assimilation, in metabolic processes of carbohydrate, amino acid and fatty acids, and in transport of carbohydrate, nitrate and proline. In terms of the cellular components, many DEGs were related to membrane, chloroplast and mitochondria, implicating a disturbance in membrane composition and energy metabolism. A total of 112 DEGs were assigned to "sequence-specific DNA binding transcription factor activity", which suggested that many transcription factors functioned in the cold response at this time point.

### Phenotypic and physiological changes in response to cold

When Manila grass was transferred to a 4°C-chamber for 2h, leaves, especially young leaves, rolled inward and remained rolled during prolonged cold stress (72 h) (Fig 6). The changes of four physiological indicators were shown in Fig 7. Water content in the leaves was reduced

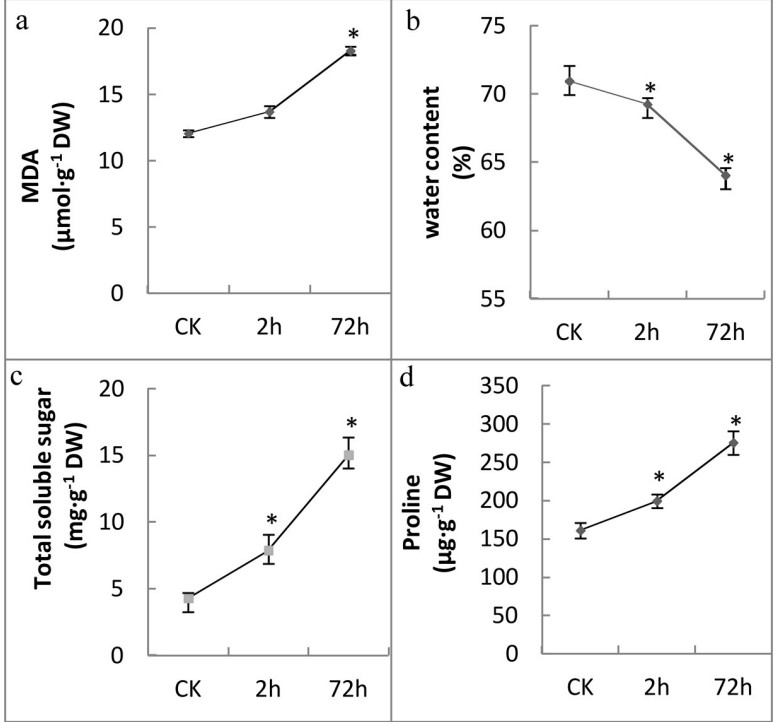

**Fig 7. Physiological changes in leaves under cold stress.** Error bars indicate SE ($n = 5$ biological replicates, $^*P<0.05$).

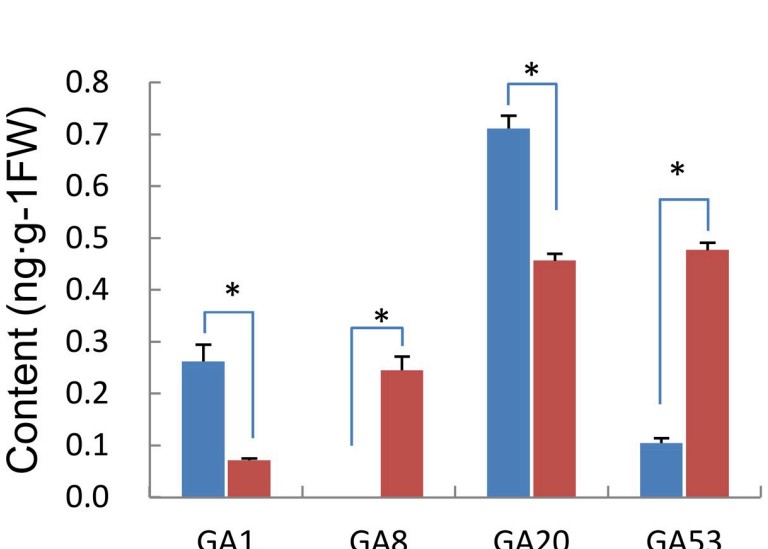

**Fig 8. GA content in leaves under cold stress.** Error bars indicate SE ($n$ = 3 biological replicates, $^*$P<0.05).

from 72.34% in the control to 69.24% and 64.04% in the 2h- and 72h-cold time points, respectively. Assava (*Manihot esculenta*), a tropical crop, also showed visible morphological changes, including the wilting of apical buds and leaves as well as downward leaves, after being exposed to 7°C-cold for 4h [36]. Thus, water balance may be a key feature of cold susceptible plants. Levels of free proline and total soluble sugars were moderately higher in the 2h-cold exposed leaves, and were significantly higher after a 72h-cold exposure. Malondialdehyde (MDA), one of the most prevalent byproducts of lipid peroxidation during oxidative stress, was found to be significantly accumulated at the 72h-cold time point, compared to the control. In addition, compared with the control, the 72h-cold leaves have a much lower levels of GA1 and GA20, while they had much higher levels of GA8 and GA53 (Fig 8). These changes indicated that cold stress resulted in drought and oxidative stresses and reduced bioactive GA levels in Manila grass.

## Discussion

### Low temperature sensing and signaling in Manila grass

Calcium is a ubiquitous and essential secondary messenger in plants, and compelling evidence indicates that calcium signaling is involved in the transduction of cold stress [37, 38]. Here we found that $Ca^{2+}$-mediated signaling also plays important roles in the cold response in Manila grass (S8 Table), especially at the 72h-cold time point. Seven genes related to $Ca^{2+}$-transport were significantly up- or down-regulated, three of which were annotated as calcium-transporting ATPase ($Ca^{2+}$ pumps), suggesting large changes in the calcium level in the cytosol. Accordingly, components of the calcium signaling were significantly modulated, most of which were up-regulated. In addition, the expressions of 10 genes encoding proteins that were predicted to be calcium-ion binding and assigned to signal transduction mechanisms by eggNOG_class_annotation were also modulated greatly. There are few reports of CBL-interacting protein kinase (CIPK) being involved in the cold stress response to date, though a number of CIPKs have been shown to play crucial roles in the regulation of stress signaling [39]. In the present

study, the expression of 16 unigenes annotated as CIPK or CIPK-like were different significantly after a 72h-cold exposure, among which CIPK2, CIPK6, CIPK7, CIPK16, CIPK19, CIPK21, CIPK22, CIPK23, CIPK24, and CIPK31 were included and were up-regulated. Similarly, 20 out of 37 detected CIPK genes in maize, including CIPK2, 7, 16, 17 and 22–25 were induced by cold [40], and in rice, 9 genes including CIPK3, 7, 9, 14, 19, 24, 26, 27 and 29 were induced by cold [41]. Ectopic expression of the BdCIPK31, a CIPK gene in *Brachypodium distachyon* that participates in low-temperature response, conferred cold tolerance in transgenic tobacco by improving ROS detoxication, omsoprotectant biosynthesis and up-regulating the expression of some representative stress-related genes under cold stress [42]. These phenomena suggest the important roles of CIPK in the cold response.

Transcription factors (TFs) are considered to play important roles in the cold response by regulating gene expression networks [4, 7]. In the present study, 156 unigenes belonging to TFs including AP2/EREBP, MYB, NAC, WRKY, bHLH, HSF and MYB-related were identified as cold regulated genes (S9 Table), suggesting their important roles in the cold response in Manila grass. The ICE-DREB-COR pathway has a major role in the cold response and has been well-characterized [13]. In our study, the transcript of an *ICE1* ortholog was induced significantly at the 72h-cold point. In *Pyrus ussuriensis* [43] and *Poncirus trifoliate* [44], the transcript level of one *ICE1* was also considerably induced by cold treatment. But in Arabidopsis and some other plants *ICEs* have been reported to be constitutively expressed [45–46]. We also identified five DREB1/CBFs genes in Manila grass. They were significantly up-regulated at the 2h-cold time point and recovered at the 72h-cold time, which was consist with expression profile of the most orthologs [47–48]. The DREB1/CBF family regulates a large spectrum of COR genes, collectively called the CBF regulon. We found that two genes annotated as 'cold acclimation protein COR413-PM1' and one gene annotated as 'dehydrin COR410' respectively were up-regulated at the 72h-cold point, but one chloroplastic COR413 inner membrane protein (COR413IM1) was down-regulated. Both COR413-PM [49] and dehydrin COR410 proteins [50–51] are potentially targeted to the plasma membrane and are up-regulated under cold or dehydration conditions. COR413IM proteins are targeted to the chloroplast envelope and cold inducible, but the activity in freezing tolerance remains to be established [52]. Homologs of other CBF regulon genes, such as COR15a and COR15b, were not detected in the present study. Similarly, the homologous genes of the Arabidopsis CBF regulon, including COR78 (At5g52310), RD29B (At5g52300), and COR6.6 (At5g15970), were not part of the tomato CBF regulon [53]. Further, in the cold sensitive plant *Ocimum americanum* var. pilosum, eight expressed unigenes were similar to those of known COR genes in Arabidopsis, including COR15A, COR47, and KIN1; none of them was substantially induced by chilling treatment [54]. Therefore, the well-established ICE-CBF-COR cold response pathway might functionally act in Manila grass, but its CBF regulon is different from that of cold tolerant plants, while it is similar to that of cold sensitive plants.

### Effect on photosystem and photosynthesis

Photosynthesis is widely assumed to be the main target of cold stress, so it is an important focus of study in the response of plants to low temperature [55, 56]. A previous report indicated that photosynthesis in zoysiagrass decreases significantly under chilling stress [57]. Our study indicated that 4°C-cold stress for 72h induced a major change in expression of genes involved in photosynthesis including light harvesting, carbon fixation and photorespiration (S10 Table).

In regard to the light harvesting complex, three and nine DGEs associated with chlorophyll biosynthesis and chlorophyll a-b binding, respectively, were down-regulated. Meanwhile, six

DEGs relating to chlorophyll catabolism were up-regulated. Accordingly, almost all DEGs relating to PSI and PSII reaction centers, and oxygen-evolving enhancer, were down-regulated. Similar gene expression changes are also seen in the *Z. japonica* [33] and other cold sensitive plants such as maize [2] and bermudagrass [18]. These findings are consistent with the observed reduction of maximal quantum yield of PS II, Fv/Fm effective quantum yield of PS II and down-regulated expression of chloroplast-related genes in maize at low temperature [2].

Carbon fixation in C4 plants includes two processes: (1) a primary fixation of $CO_2$ by the carboxylation of phosphoenolpyruvate (PEP) in the mesophyll cells, and (2) the reduction of $CO_2$ to carbohydrate via the Calvin cycle within the bundle sheath cells. PEP carboxylase (PEPC) is a key enzyme in primary $CO_2$ fixation, and chloroplastic phosphoribulokinase and ribulose bisphosphate carboxylase (Rubisco) are the two key proteins involved in the Calvin cycle. After a 72h-cold exposure, three DEGs annotated as carbonic anhydrase were down-regulated, suggesting that the supply of $CO_2$ is reduced in the cold. At the same time, two DEGs annotated as chloroplastic Rubisco large subunit-binding protein subunit beta and chloroplastic Rubisco small chain respectively, and one DEG annotated as phosphoribulokinase were down-regulated. The expression change of these genes supports a reduction of photosynthesis in cold conditions [57]. It has been shown before that the activity of PEPC [58] in zoysiagrass decreased significantly under chilling stress, compared to that of plants grown at 25˚C. According to our transcriptome data, 15 unigenes annotated as PEPC were identified; four of them were identified as DEGs and were all up-regulated. Therefore, low PEPC activity in chilling conditions should be independent of transcription changes.

The inhibition of $CO_2$ assimilation will lead to an energy excess that was indicated as the over-reduction of the photosynthetic electron transport chain [54]. Three DEGs annotated as chloroplastic ferredoxin-NADP reductase (FNR), chloroplastic rhodanese domain-containing protein (RDCP) and chloroplast NAD(P)H dehydrogenase (NDH) respectively, were up-regulated at the 72h-cold point. Chloroplastic FNR catalyzes the final step of the photosynthetic electron transport chain, namely, electron transfer from reduced Fd to $NADP^+$ [59], and RDCP is required for anchoring FNR to the thylakoid membranes and sustaining efficient linear electron flow [60]. The chloroplast NDH complex shuttles electrons from NAD(P)H to plastoquinone, and thus conserves the redox energy in a proton gradient, which has been shown to alleviate oxidative damage [56, 61]. The up-regulation of the three genes indicated that photosynthetic electron transport chain was overloaded. The photorespiratory pathway consumes excess energy and reductive equivalents, and contributes to avoidance of photoinhibition [62, 63]. Usually, photorespiratory activity is enhanced when plants are under stress. In the present study, we found that six DEGs encoding enzymes involved in photorespiration including peroxisomal (S)-2-hydroxy-acid oxidase, glycine cleavage and hydroxypyruvate reductase were all down-regulated, suggesting the weak activity of photorespiration in Manila grass under cold stress.

Taken together, the transcriptome data suggested that cold stress induces a rearrangement of the photosynthetic apparatus, inhibition of carbon fixation, reduction in photorespiration and over-reduction of the photosynthetic electron transport chain in Manila grass.

## Effect on nitrogen metabolism

Nitrogen is an essential macronutrient and a key factor limiting plant growth. In leaf cells, cytoplasmic ammonium pools are replenished not only by nitrate/nitrite reduction or ammonium uptake, but also by ammonium released from internal metabolism including photorespiration, lignin biosynthesis and protein turnover [64]. Either exogenous or internal ammonium in cells is primarily assimilated into the plastid/chloroplast by the so-called glutamine synthase

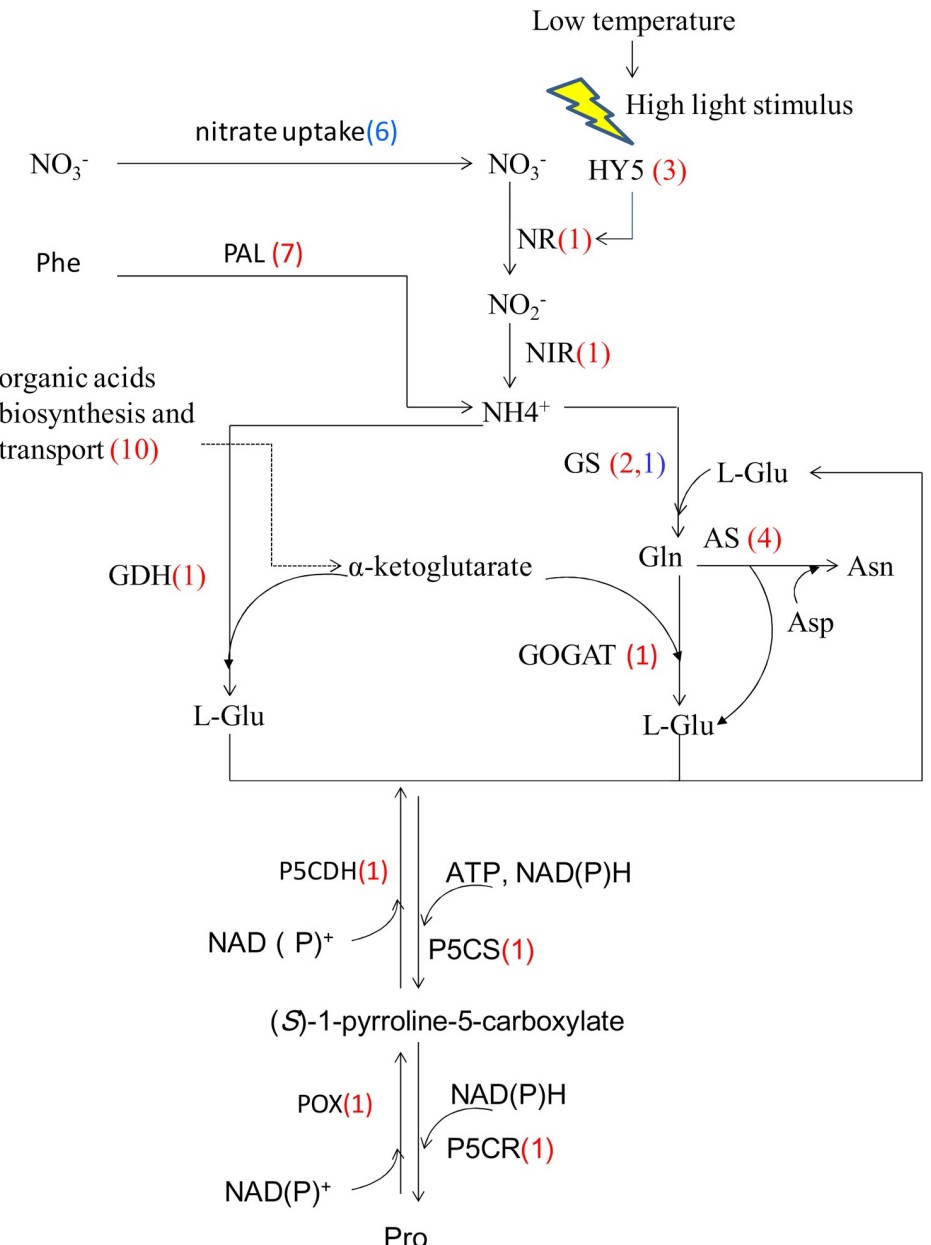

**Fig 9. DEGs involved in nitrogen assimilation.** Phe: phenylalanine; PAL:phenylalanine ammonia-lyase; HY5: Transcription factor long hypocotyl 5; NR: Nitrate reductase; NIR: Ferredoxin—nitrite reductase; GS: Glutamine synthetase; AS:Asparagine synthetase; GOGAT: glutamate synthase; GDH: Glutamate dehydrogenase; P5CS:Delta 1-pyrroline-5-carboxylate synthetase; P5CR: Pyrroline-5-carboxylate reductase; POX: Proline dehydrogenase; P5CDH: Delta-1-pyrroline-5-carboxylate dehydrogenase. In brackets, red numbers represent the number of up-regulated unigenes, blue numbers represent the number of down-regulated unigenes, the same below.

/ glutamate synthase (GS/GOGAT) cycle [65]. Manila grass exposed to cold stress for 72h modulated the expression of multiple genes involved in nitrogen metabolism (Fig 9, S11 Table).

At the 72h-cold point, six DEGs involved in the uptake of nitrate, including five of NRT1/ PTR (nitrate transporter 1/peptide transporter) family and one of chloride channel (CLC) family, were down-regulated. A study of Arabidopsis indicated that the expression of NRT1.5 was

reduced so as to cut down the nitrate assimilation and save energy, which functions to promote stress tolerance [66]. In contrast to these findings, one DEG annotated as "probable nitrite transporter At1g68570-like" (AtNPF3.1) was up-regulated, while one annotated as PII protein (Piriformospora indica-insensitive protein 2) was down regulated. In Arabidopsis the two genes are a single homolog. AtNPF3.1 is a chloroplastic nitrite transporter and functions in loading cytosolic nitrite into chloroplasts [67]. A more recent study indicated that AtNPF3.1 gene expression was up-regulated by low exogenous nitrate concentrations [68]. In *Chlamydomonas* the chloroplastic nitrite transporter was found to be critical for cells' survival under limiting nitrate conditions [69]. PII protein is involved in the down-regulation of $NO_2^-$ uptake into Arabidopsis rosette leaf chloroplasts in the light. The PII knock-out mutants showed an increased light-dependent nitrite uptake into chloroplasts when compared to the wild-type [70]. The up-regulated expression of AtNPF3.1 and the down regulation of PII protein suggested that nitrite uptake into chloroplasts is stimulated. Compared to the reduced uptake of exogenous $NO_3^-$, the internal ammonium supply might be enhanced. Phenylalanine ammonia-lyase (PAL) catalyzes the reaction: L-phenylalanine → trans-cinnamate + ammonia. All eight DEGs annotated as PAL were up-regulated, which suggested an increased degradation of phenylalanine under cold stress. Corresponding to this, two DEGs annotated as ammonium transporter AMT4;1 and AMT2;1 respectively, were up-regulated, while another transporter gene AMT3;3 was down-regulated. In Arabidopsis, AtAMT2 is allocated to the plasma membrane, mainly mediating ammonium import [71]; the function of AMT3 and AMT4 homologs are largely unknown. The expression changes of genes mentioned above implied that the uptake of exogenous nitrate was reduced, and the transport of nitrite into the chloroplast as well as the import of ammonium into cells was stimulated in order to supply enough nitrogen.

Although the import of apoplastic nitrate is reduced, the nitrogen assimilation in cells was stimulated, as indicated by the up-regulation of 10 genes encoding enzymes involved in the process, including NR (nitrate reductase), NiR (nitrite reductase), GS root isozyme, chloroplastic ferredoxin-dependent GOGAT, glutamate dehydrogenase (GDH), asparagine synthase (AS) and aspartate aminotransferase. Proline, acting as an osmoregulator and cellular protectant, is always accumulated in plants under multiple abiotic stresses including drought, cold and high salinity. In general, proline biosynthesis through the glutamate pathway is predominant under stress conditions, in which pyrroline-5-carboxylate synthase (P5CS) and P5C reductase (P5CR) are the two key enzymes. In the present study, two DEGs annotated as P5CS and P5CR respectively, were up-regulated. At the same time, two DEGs encoding proline dehydrogenase and P5C dehydrogenase (P5CD) respectively, that are involved in the proline catabolic process, were also up-regulated, indicating that both the biosynthesis and degradation of proline were activated in Manila grass by cold stress. The higher proline levels in leaves at the 72h-cold point relative to the control suggested that the biosynthetic process was dominant over the degradation process. Stimulated nitrogen assimilation might be important to providing glutamate for the biosynthesis of proline.

Cellular carbon and nitrogen metabolism are tightly coordinated [72]. Nitrogen assimilation affects carbon metabolism, as indicated by the up-regulation of the transcription of genes involved in the biosynthesis of organic acids [73]. In leaves of Manila grass exposed to cold for 72h, a DEG annotated as glyoxysomal malate synthase, a key enzyme in the glyoxylate cycle, and several DEGs encoding enzymes in the tricarboxylic acid cycle, including aconitate hydratase, NAD-malate dehydrogenase and 2-oxoglutarate dehydrogenase complex, were up-regulated. These expression changes will promote the production of organic acids needed in the synthesis of amino acids. Corresponding to this, three and one DEGs annotated as tonoplast dicarboxylate transporter (TDT) and aluminium-activated malate transporter-10 (ALMT-10),

respectively, were up-regulated. TDTs are allocated to the tonoplast membrane and are critical for the regulation of pH homeostasis by transporting malate into vacuoles. A positive correlation between acidification of the incubation medium and the accumulation of mRNA of a TDT gene was observed in Arabidopsis leave discs [74]. ALMT proteins are located in either the plasma membrane or tonoplast, being regulated by $Al^{3+}$, pH changes, phosphorous concentration, thus allowing malate to flow out of the cell or into the vacuole [75]. The enhanced expression of genes relating to organic acid biosynthesis and transport further support the contention that nitrogen assimilation is stimulated by cold.

Collectively, the modulation in the expression of genes involved in nitrogen metabolism suggest that cold stress reduce the uptake of apoplastic nitrate but stimulate the nitrogen assimilation in cells. Maintaining an appropriate balance of carbohydrates to nitrogen metabolites will be important for Manila grass to survive cold stress.

**Effect on carbohydrate metabolism.** Starch and sucrose are the primary products of photosynthesis and energy resources in higher plants. Starch degradation is considered to produce osmoprotectants, free radical scavengers, and stabilizers of protein and cellular structure under multiple abiotic stresses [76]. Changes in expression of many genes involved in the metabolism and transport of starch and sucrose were also observed in Manila grass exposed to cold (Fig 10, S12 Table). Both α-amylase activity (GO:0004556) and β-amylase activity (GO:0016161) were enriched in the DEGs between 72h_ CT VS CK (S5 Table). Among these DEGs, three genes annotated as α-amylase and β-amylase were up-regulated, indicating that starch degradation was stimulated by cold. In addition, one, one, and two DEGs annotated as hexokinase, fructokinase and sucrose synthase, respectively, were up-regulated. Accordingly,

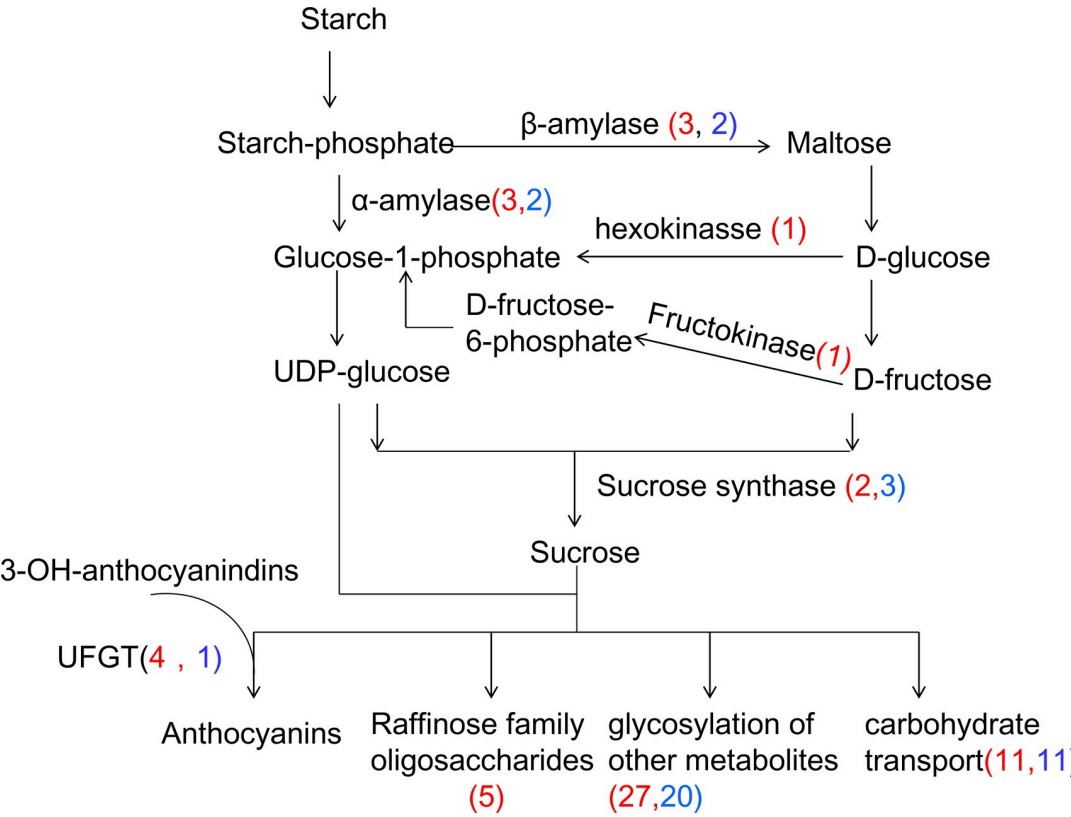

**Fig 10. Effect of cold on starch and sucrose metabolism.** UFGT:anthocyanidin 3-O-glucosyltransferase.

three GO terms including "response to sucrose" (GO:0009744), "response to glucose" (GO:0009749) and "response to fructose" (GO:0009750) were enriched among the DEGs, suggesting the accumulation of these sugars. Raffinose family oligosaccharides (RFOs) are α-1, 6-galactosyl extensions of sucrose [77] and have been reported to accumulate significantly under low temperature stress [78–80]. Stachyose synthase, galactinol synthase and galactinol-sucrose galactosyltransferase are important for the biosynthesis of RFOs. Five DEGs predicted to encode these enzymes were up-regulated significantly, suggesting the accumulation of RFO in Manila grass in cold conditions (S12 Table). At the physiological level, the concentration of total soluble sugars at this cold time point was much higher when compared to the control.

The accumulation of soluble sugars stimulated sugar transportation and glycosylation of metabolites, which was indicated by the enrichment of GO terms of "sugar transmembrane transporter activity", "carbohydrate transport", "transferase activity, transferring glycosyl groups" and "quercetin 3/4/7-O-glucosyltransferase" in the DEGs between the 72h-cold and control (S5 Table). A total of 22 DEGs were assigned to the term "carbohydrate transport", half of them, including one annotated as the bidirectional sugar transporter SWEET16, were up-regulated. SWEET16, acting as a vacuolar hexose transporter for sugars such as glucose, fructose, and sucrose, regulates sugar homeostasis by exporting/importing them through the tonoplast. Arabidopsis plants overexpressing SWEET16 showed a substantially modified sugar composition in the cold and exhibited strongly improved freezing tolerance when compared with wild-type plants [81]. A total of 44 DEGs were assigned to the term "transferring glycosyl groups", among which 12 DEGs were predicted to show quercetin O-glucosyltransferase activity. The biosynthesis of anthocyanins is a process which involves glycosylation of metabolites. A good correlation between sugar content and anthocyanin accumulation has been observed in various plant species [82–83]. Induction of anthocyanin accumulation by low temperature has also been reported in leaves of maize [84], Arabidopsis [85] and in fruits of orange [86] and pear [87]. In the present study, three and one DEGs annotated as anthocyanidin 5, 3-O-glucosyltransferase (A5,3GT) and anthocyanidin 3-O-glucosyltransferase (A3GT), respectively, were up-regulated, suggesting induced anthocyanin accumulation in Manila grass by cold.

**Changes in plant hormones metabolism and signaling.** It is generally accepted that hormone homeostasis plays an important role in plant responses to abiotic stresses by mediating growth, development, nutrient allocation, and source/sink transitions. Abscisic acid (ABA) is considered as a stress-responsive hormone and has been studied deeply. Besides this, the role of cytokinin (CK), gibberellin (GA), and auxin during environmental stress is emerging [88]. Our transcriptome data indicated that the metabolism and signaling of these hormones were modulated significantly in Manila grass exposed to cold (S13 Table), of which GA and CK deserve attention especially.

Gibberellin is an important plant hormone. It not only promotes processes of plant growth and development, such as seed germination, cell elongation and fruit development, but also regulates the adaptation to environmental [89]. The levels of bioactive GAs in plants are maintained by the regulation of GA biosynthesis and catabolism. In the present work, nine unigenes assigned to GO:0009685 (gibberellin metabolic process) and GO:0045487 (gibberellin catabolic process) were over-represented in DEGs between CT_72 VS CK (S13 Table). The expressions of one CPS1 (ent-copalyl diphosphate synthase 1 and one A20ox (GA20-oxidase) unigene were down regulated, while four GA2ox (gibberellin 2-oxidase) unigenes were up-regulated. CPS1 catalyzes the synthesis of gibberellin precursor ent-copalyl diphosphatec from geranylgeranyl diphosphate [90]. GA20ox catalyzes the removal of the C-20 of GA12 in the formation of C19-GAs, which is a key step in the GA biosynthetic pathway [91]. GA2ox is responsible for catabolism of precursors of the active GAs, GA20 and GA9 to their inactive

catabolite forms, GA29 and GA51, respectively, and catabolism of active GA1 and GA4 to their inactive catabolite forms, GA8 and GA34, respectively. The synchronous down-regulation of *CPS1* and *A20ox* plus the up-regulation of *GA2ox* suggest that active GA levels decreased by both reducing GA biosynthesis and enhancing GA catabolism (Fig 11). At physiological level, compared to the control, the 72h-cold treatment showed lower levels of GA1 and GA20 while higher levels of GA8 and GA53, which is consistent with the changes observed

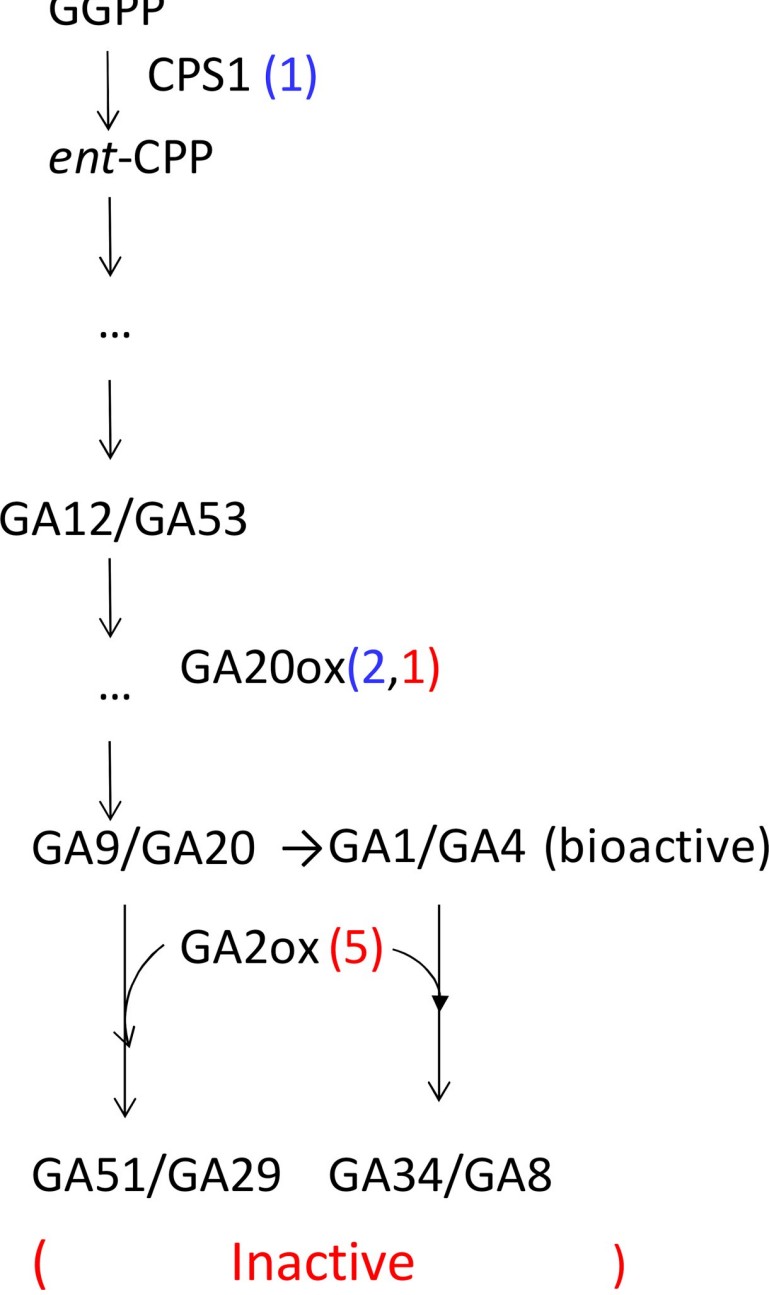

**Fig 11. Effect of cold on gibberellin biosynthesis and catabolism.** GGPP: (E, E, E)-geranylgeranyl diphosphate; CPP: *ent*-copalyl diphosphate; CPS1: *ent*-copalyl diphosphate synthase1; GA20ox: GA20-oxidase; GA2ox: gibberellin 2-oxidase.

at transcription level. Previous findings in wheat (*Triticum aestivum*) [92], *Dendranthema grandiflorum* [93] and sunflower (*Helianthus annuus*) [94] also indicate that endogenous GA levels decrease as temperature is lowered and this is associated with decreased shoot growth. In the GA signaling pathway, we observed that two DEGs at the 72h-cold point were annotated as gibberellin receptor GID1L2 and GID1-like respectively, indicating that the cold induced changes in GA levels might act on both GID1L2 and GID1 signaling (S13 Table). GID1L2 signaling was also triggered in the growth-limited leaves of *Brachypodium distachyon* under drought stress [95]. The significant expression changes of three unigenes annotated as gibberellin-responsive protein were also observed, indicating that GA signaling was modulated.

Cytokinin (CK) play crucial roles throughout a plant's life span [96]. In the present study, 13 unigenes involved in cytokinin metabolism were included among the DEGs between 72h_CT VS CK (Fig 12, S13 Table). Among the DEGs, seven were annotated as cytokinin riboside 5'-monophosphate phosphoribohydrolase (LOG) and six of them were down regulated. In addition to this, the expression of three and one DEGs annotated as cytokinin glucosyltransferase (CKG) and cytokinin oxidase/dehydrogenase (CKX) respectively was modulated greatly. LOG is a cytokinin-activating enzyme, playing a pivotal role in regulating cytokinin activity [97]. CKG and CKX catalyze the deactivation of CKs by glycosylation and degradation, respectively [98–99]. The down-regulation of LOG genes plus the up-regulation of CKG and CKX in the cold would lead to a reduction in the level of bioactive cytokinins (Fig 12). A total

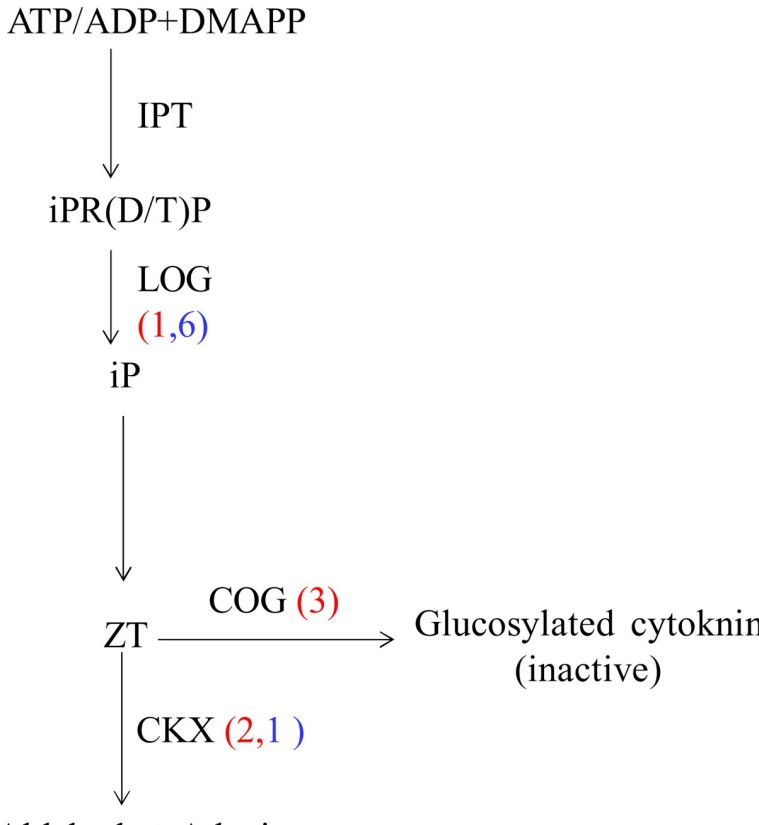

**Fig 12. Effect of cold on cytokinin biosynthesis and catabolism.** DMAPP: dimethylallyl pyrophosphate; iP: isopentenyladenine; ZT: zeatin; IPT: isopentenyl transferases; LOG: Cytokinin riboside 5'-monophosphate phosphoribohydrolase; CKX: cytokinin dehydrogenase; COG: Cytokinin-O-glucosyltransferase.

of eight DEGs were assigned to the term "response to cytokinin", indicating the impact on the cytokinin signal pathway.

GA and CK mainly participate in the regulation of growth and development. The reduction of bioactive style of the hormones should be related to the growth retardation and senescence of Manila grass in cold condition.

## Cold stress is coupled with multiple stresses

Cold stress is always coupled with high light stress [100]. In the present study, the terms GO:0009416 (response to light stimulus) and GO:0009644 (response to high light intensity) were enriched among DEGs between CT_72h VS CK. The light stress-induced expression of ELIPs (early light-inducible proteins) in green plants increases with the light intensity and the duration of the stress [101]. ELIPs are thought to protect plastids against light stress [102]. In Manila grass exposed to cold for 72h, two unigenes annotated as ELIPs were up-regulated with fold changes of 11.08 and 8.90 (S14 Table), indicating that the plant is subjected to high light stress. Transcription factor long hypocotyl 5 (HY5) and HYH (an HY5 homolog) are high light induced and activators of nitrate reductase [46,103–104]. The expression of NR is also induced by light [105]. At this cold point, three and two DEGs annotated as HY5 and NR respectively were up-regulated (S14 Table). Excess light can lead to free-radical formation and photooxidation processes, which is harmful to the photosynthetic apparatus.

It has been well established that low temperature can perturb the electron transport chain and metabolic processes, promoting the production of reactive oxygen species [93, 106]. After a 2h-cold exposure, the GO term "response to oxidative stress" (GO:0006979) was enriched among the DEGs; at the 72h cold-time point, 238 DEGs genes were involved in "oxidation-reduction process" (GO:0055114). Accordingly, GO terms "response to oxygen-containing compound" (GO:1901700) and "response to singlet oxygen" (GO:0000304) were enriched among the DEGs. The MDA level was found to be significantly higher after a cold treatment compared with the control (Fig 8). These data indicate that oxidative stress was induced by cold and that damage to the cell membrane occurred. Catalase (CAT), superoxide dismutase (SOD) and peroxidase (POD) play important roles in the scavenging of ROS. In the present study, 10, 15 and 170 unigenes annotated as CAT, SOD and POD respectively, were identified. At the 2h_cold point, only four POD genes were identified as DEGs, of which two were up-regulated while two were down-regulated (S14 Table). At the 72h-cold point, three POD unigenes were up-regulated, while 11 POD genes and two SOD genes were down-regulated (S14 Table); none of the CAT genes were responsive to cold. Physiological detection indicated that CAT, SOD and POD activities were not changed significantly between cold treatments and the control. Reduced glutathione (GSH) is a critical molecule in resisting oxidative stress and maintaining the reducing environment of cells. At the 72h-cold point, 35 DEGs were assigned to the GO term "glutathione metabolism" (S7 Table), among them one gene annotated as glutathione-disulfide reductase (GDR) and 16 genes annotated as glutathione S-transferase (GST) were up-regulated (S14 Table). GDR catalyzes the reduction of glutathione disulfide (GSSG) to GSH, GSTs catalyze the transfer of the GSH to a cosubstrate (R-X) containing a reactive electrophilic center to form a polar *S*-glutathionylated reaction product (R-SG). The expression changes of these genes suggest that, in Manila grass under cold stress, the activity of CAT, POD and SOD were not triggered, while GSH may be important in the alleviation of oxidative stress.

In cold-sensitive plants, low temperature conditions also cause leaf dehydration [107]. The rolling leaves and reduced water content induced by cold stress indicated the grass was drought stressed. This presence of drought stress was also supported by the following pieces of

evidence. First, GO term "response to water deprivation" (GO:0009414) was enriched, to which 45 DEGs were assigned, among them 37 were up-regulated (S14 Table). Secondly, levels of free proline and soluble sugars were significantly higher in the 72h-cold exposed leaves, compared with the control (Fig 8). It is well known that free proline and soluble sugars are involved in osmoregulation, stabilizing of sub-cellular structures and scavenging free radicals. They are accumulated in higher plants under a number of abiotic stresses including low temperature, drought and high salinity. Finally, three, nine, nine and 25 DEGs annotated as dehydrin (DHN), heat shock protein (HSP), heat shock factor (HSF) and late embryogenesis abundant (LEA) protein, respectively, were up-regulated significantly. DHNs, HSPs and LEA proteins are quite hydrophilic and function in the direct protection of the cell from stress by increasing membrane stability, preventing incorrect folding and processing of proteins and by other still unclear mechanism [108–109]. HSFs are the transcription factors that regulate the expression of the HSPs. The up-regulation of these genes is considered as an indicator of low temperature and drought stress [110–111]. These data suggested that Manila grass suffered cold coupled with desiccation stress.

The multiple stresses would accelerate the senescence of Manila grass in cold environment.

## Conclusions

To sum up, our study presented the first comprehensive transcriptome data of *Z. matrella* leaves using Illumina sequencing technology. A total of 82,605 unigenes were obtained with the Trinity *de novo* assembly method. The large number of transcripts identified will serve as a global resource for future research. The results of this large-scale gene expression survey identified 6,175 cold-responding unigenes. Overall, cold stress reduced the expression of a number of genes related to the photosynthetic apparatus, light harvesting reaction, and carbon fixation, and induced the expression of some key genes in nitrogen assimilation, in addition to some genes that function in the reduction of the levels of bioactive CK and GA. Cold stress also triggered changes in the expression of many genes involved in oxidation-reduction reactions, amino acid metabolism and carbohydrate metabolism. Calcium signaling and the ICE-CBF-COR pathway are part of cold signal sensing and transduction. Manila grass under cold stress was coupled with high light, oxidative and drought stresses, but the expression of antioxidant enzymes including CAT, SOD, and POD, were only weakly induced by cold. In turf management, often attempts are made to lessen cold damage by irrigation and supplying of nitrogenous fertilizer before a period of cold temperatures. Further, in breeding research, selection of strains with higher expression levels of antioxidant enzymes under cold stress is worthwhile. Finally, the regulation of plant hormones CKs and GAs in Manila grass under cold stress is worthy of further study.

## Supporting information

**S1 Table. Differentially expressed genes between 2h_CT and CK.**
(XLSX)

**S2 Table. Differentially expressed genes between 72h_CT and CK.**
(XLSX)

**S3 Table. Primers used for qRT-PCR.**
(XLSX)

**S4 Table. Enriched GO terms between 2h_CT and CK.**
(XLSX)

**S5 Table. Enriched GO terms between 72h_CT and CK.**
(XLSX)

**S6 Table. Enriched KEGG pathways between 2h_CT and CK.**
(XLSX)

**S7 Table. Enriched KEGG pathways between 72h_CT and CK.**
(XLSX)

**S8 Table. DEGs related to calcium signaling.**
(XLSX)

**S9 Table. Representative cold responding transcription factor families.**
(XLSX)

**S10 Table. DEGs related to photosystem and photosynthesis.**
(XLSX)

**S11 Table. DEGs related to nitrogen assimilation.**
(XLSX)

**S12 Table. DEGs related to starch and sucrose metabolism.**
(XLSX)

**S13 Table. DEGs related to phytohormone metabolism and signaling.**
(XLSX)

**S14 Table. DEGs related to other stresses and responses.**
(XLSX)

## Acknowledgments

We would like to thank Sarah J. Gilmour (Plant Research Lab, Michigan State University) very much for language modification.

## Author Contributions

**Conceptualization:** Zhenyuan Sun, Shanjun Wei.

**Funding acquisition:** Zhenyuan Sun, Shanjun Wei.

**Investigation:** Sixin Long, Fengying Yan, Shanjun Wei.

**Resources:** Shanjun Wei.

**Writing – original draft:** Shanjun Wei.

**Writing – review & editing:** Lin Yang, Shanjun Wei.

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
