## [Decision Letter · Decision Letter 0]

31 Mar 2020

PONE-D-20-00949

Responses of Manila Grass (Zoysia matrella) to cold stress: from transcriptomics to physiology

PLOS ONE

Dear Dr Wei,

Thank you for submitting your manuscript to PLOS ONE. After careful consideration, we feel that it has merit but does not fully meet PLOS ONE’s publication criteria as it currently stands. Therefore, we invite you to submit a revised version of the manuscript that addresses the points raised during the review process.

We would appreciate receiving your revised manuscript by May 15 2020 11:59PM. To enhance the reproducibility of your results, we recommend that if applicable you deposit your laboratory protocols in protocols.io, where a protocol can be assigned its own identifier (DOI) such that it can be cited independently in the future. For instructions see: http://journals.plos.org/plosone/s/submission-guidelines#loc-laboratory-protocols

We look forward to receiving your revised manuscript.

Kind regards,

Anil Kumar Singh, Ph.D.

Academic Editor

PLOS ONE

Journal Requirements:

https://www.ncbi.nlm.nih.gov/pubmed/26115186

https://journals.plos.org/plosone/article?id=10.1371/journal.pone.0050785

https://www.tandfonline.com/doi/abs/10.1080/07352689.2014.870411?journalCode=bpts20

https://onlinelibrary.wiley.com/doi/full/10.1111/j.1365-313X.2007.03100.x

http://www.plantphysiol.org/content/143/2/639.full

In your revision ensure you cite all your sources (including your own works), and quote or rephrase any duplicated text outside the methods section. Further consideration is dependent on these concerns being addressed.

Reviewers' comments:

Reviewer's Responses to Questions

**Comments to the Author**

1. Is the manuscript technically sound, and do the data support the conclusions?

Reviewer #1: Yes

Reviewer #2: Yes

2. Has the statistical analysis been performed appropriately and rigorously? 

Reviewer #1: Yes

Reviewer #2: Yes

3. Have the authors made all data underlying the findings in their manuscript fully available?

Reviewer #1: Yes

Reviewer #2: Yes

4. Is the manuscript presented in an intelligible fashion and written in standard English?

Reviewer #1: Yes

Reviewer #2: Yes

5. Review Comments to the Author

Reviewer #1: Study by Long et al on transcriptomic profiling of Manila grass under cold stress presents alteration in transcripts in response to cold stress and related physiological traits presented as markers of stress. Authors have presented a huge set of information in terms changes in transcript levels involve in various metabolic pathways and physiological process. Although the information presented is useful and can be utilized in related future studies, here in this study the conclusions made by authors are mostly speculative and not interconnected. I agree that 'omics' data is always huge and it is difficult to perform a directional analysis and reach to logical conclusions. A cold stress exposure would certainly change every physiological trait and metabolic pathway in the plant. The necessary part is the analysis of the data, which is entirely missing in this study. Almost everything is covered which has made the things even more complex to understand. Physiological data is very generalized and adds nothing new in the study.

I would highly recommend author to focus the manuscript in a way that can be connected to cold susceptibility or tolerance and exclude rest of the data which is not really serving the purpose. Avoid speculative conclusions as they may be misleading unless validated. Authors are also suggested to reduce the overall length of manuscript with a proper rationale, brief introduction and a more focussed discussion of findings.

Reviewer #2: In the manuscript entitled "Responses of Manila Grass (Zoysia matrella) to cold stress: from transcriptomics to

physiology", authored by Long et al., done a good piece of work, but few comments are to be answered:

1- Do manila grass wilts in the winter or it completes its life cycle?

2- Plant material and treatment: why this temperature was chosen?

3- day night hours not mentioned.

4- Too long discussion.

5- Why only leaves not roots were taken for study?

6. PLOS authors have the option to publish the peer review history of their article (what does this mean?). If published, this will include your full peer review and any attached files.

Reviewer #1: Yes: RAJEEV NAYAN BAHUGUNA

Reviewer #2: No

---

## [Author Response · Author response to Decision Letter 0]

15 May 2020

Dear reviewers and editor,

 Thank you for giving us the chance to modify the manuscript. Wei learned a lot during the modifying process. The following are the details.

 1. To academic editor

 （1）Please ensure that your manuscript meets PLOS ONE's style requirements, including those for file naming. 

 We have downloaded two FDF files: “PLOSOne_formatting_sample_main_body” and “PLOSOne_formatting_sample_title_authors_affiliations”. Our manuscript has been arranged under the guide of the two files. 

 （2） We noticed you have some minor occurrence of overlapping text with the following previous publication(s), which needs to be addressed。 

We have revised the manuscript carefully and tried our best to rewrite the overlapping text. 

 2、To Reviewer #1

 Study by Long et al on transcriptomic profiling of Manila grass under cold stress presents alteration in transcripts in response to cold stress and related physiological traits presented as markers of stress. Authors have presented a huge set of information in terms changes in transcript levels involve in various metabolic pathways and physiological process. Although the information presented is useful and can be utilized in related future studies, here in this study the conclusions made by authors are mostly speculative and not interconnected. I agree that 'omics' data is always huge and it is difficult to perform a directional analysis and reach to logical conclusions. A cold stress exposure would certainly change every physiological trait and metabolic pathway in the plant. The necessary part is the analysis of the data, which is entirely missing in this study. Almost everything is covered which has made the things even more complex to understand. Physiological data is very generalized and adds nothing new in the study.

I would highly recommend author to focus the manuscript in a way that can be connected to cold susceptibility or tolerance and exclude rest of the data which is not really serving the purpose. Avoid speculative conclusions as they may be misleading unless validated. Authors are also suggested to reduce the overall length of manuscript with a proper rationale, brief introduction and a more focussed discussion of findings.

Response: Since the transcriptome data is huge and complex. Many clues can be available, but it is not easy to draw firm conclusions, unless corresponding physiological data are also available. In this study, five physiological characters were detected to assist the analysis of the transcriptome data. Among the five physiological parameters, the desecration stress and low bioactive GA level induced by cold are first reported in Manila grass, although they have been reported in other species. Because of the character of transcriptome data, only the direct results were provided in the section of results, most of analysis is arranged in the discussion section. Basing on our transcriptome data as well as the physiology data of our own and of documents, the cold-responding characters in Manila grass were discussed from five aspects: cold sensing and signaling, changes on system and photosynthesis activity, effect on nitrogen metabolism and carbohydrates, metabolism and signaling of plant hormones, and stresses coupled with cold. According to the comment, we read over the manuscript seriously and revised it carefully, especially in the discussion section. In details, in the first aspect, words that interpret the calcium pathway and transcription factors have been reduced; In the second aspect, the activity of mitochondria on photosynthesis has been deleted and the remained content has been reorganized; In the aspect of nitrogen metabolism, words that interpret the transport of nitrate and the assimilation process haves been reduced; in the aspect of plant hormones, only content of GA and CK that could come clear conclusions were remained; In the aspect of multiple stresses, the first paragraph as well the content related to karrikins have been deleted. The revised discussion seems more readable, though there are still many deficiencies need to improve on academic aspect. 

 3. To Reviewer #2

 (1) Do manila grass wilts in the winter or it completes its life cycle?

Response: Manila grass is a type of perennial turfgrass. In winter, the shoots wilts and the stolons are dormant. But if it is freezing severely, the dormant stolons will also die off. The word “perennial” has been supplemented in the introduction as following: For instance, Manila grass (Zoysia matrella Merr.), a popular warm season perennial turfgrass, shows excellent tolerance to drought and high temperature stresses, but is sensitive to low temperature. The turf it builds always exhibits a long yellow period in temperate zones or even dies off during winter freezes, which compromises its ornamental value and usage greatly.

 (2) Plant material and treatment: why this temperature was chosen?

 Response: 4℃-cold is often used to treat plant for cold-responding study. So we also used the condition to investigate the cold response in Manila grass. 

 (3)day night hours not mentioned.

 Response: The day/night hours are 14h/10h. In the method section, we say: For cold treatment, plants were transferred to a growth chamber set to a 14h-photoperiod. 

 (4)Too long discussion.

 Response:The discussion section has been reduced, as described in the response to reviewer #1. 

 (5)Why only leaves not roots were taken for study?

 Response: Since leaves are sensitive to low temperature. When the atmospheric temperature drops to 10℃ or lower, leaves turn yellow and wilt gradually.

---

## [Decision Letter · Decision Letter 1]

18 Jun 2020

PONE-D-20-00949R1

Responses of Manila Grass (Zoysia matrella) to cold stress: from transcriptomics to physiology

PLOS ONE

Dear Dr. Wei,

Thank you for submitting your manuscript to PLOS ONE. After careful consideration, we feel that it has merit but does not fully meet PLOS ONE’s publication criteria as it currently stands. Therefore, we invite you to submit a revised version of the manuscript that addresses the points raised during the review process.

Authors are advised to clarify the concern raised by the reviewer 2 and make the necessary changes accordingly.

We look forward to receiving your revised manuscript.

Kind regards,

Anil Kumar Singh, Ph.D.

Academic Editor

PLOS ONE

Reviewers' comments:

Reviewer's Responses to Questions

**Comments to the Author**

1. If the authors have adequately addressed your comments raised in a previous round of review and you feel that this manuscript is now acceptable for publication, you may indicate that here to bypass the “Comments to the Author” section, enter your conflict of interest statement in the “Confidential to Editor” section, and submit your "Accept" recommendation.

Reviewer #1: All comments have been addressed

Reviewer #2: All comments have been addressed

2. Is the manuscript technically sound, and do the data support the conclusions?

Reviewer #1: Yes

Reviewer #2: Yes

3. Has the statistical analysis been performed appropriately and rigorously? 

Reviewer #1: Yes

Reviewer #2: Yes

4. Have the authors made all data underlying the findings in their manuscript fully available?

Reviewer #1: Yes

Reviewer #2: Yes

5. Is the manuscript presented in an intelligible fashion and written in standard English?

Reviewer #1: Yes

Reviewer #2: Yes

6. Review Comments to the Author

Reviewer #1: (No Response)

Reviewer #2: One point is that there is quite difference in cold stress, chilling stress and frost stress. Please check which one fits in your criteria as in grass species all three affects differentially, and change the title accordingly.

7. PLOS authors have the option to publish the peer review history of their article (what does this mean?). If published, this will include your full peer review and any attached files.

Reviewer #1: Yes: RAJEEV NAYAN BAHUGUNA

Reviewer #2: No

---

## [Author Response · Author response to Decision Letter 1]

24 Jun 2020

Cold stress includes chilling stress (0 - 20 °C) and freezing stress (< 0 °C) (Chinnusamy et al., 2007). Strictly speaking, we should use chilling stress instead of cold stress. Therefore, we has changed the tile to “Responses of Manila Grass (Zoysia matrella) to chilling stress: from transcriptomics to physiology”. 

Chinnusamy, V., Zhu, J., & Zhu, J.-K. (2007). Cold stress regulation of gene expression in plants. Trends in Plant Science, 12(10), 444–451. doi:10.1016/j.tplants.2007.07.002

---

## [Editor Report · Decision Letter 2]

26 Jun 2020

Responses of Manila Grass (Zoysia matrella) to chilling stress: from transcriptomics to physiology

PONE-D-20-00949R2

Dear Dr. Wei,

We’re pleased to inform you that your manuscript has been judged scientifically suitable for publication and will be formally accepted for publication once it meets all outstanding technical requirements.

Kind regards,

Anil Kumar Singh, Ph.D.

Academic Editor

PLOS ONE
---

## [Editor Report · Acceptance letter]

6 Jul 2020

PONE-D-20-00949R2 

Responses of Manila Grass (*Zoysia matrella*) to chilling stress: from transcriptomics to physiology 

Dear Dr. Wei:

I'm pleased to inform you that your manuscript has been deemed suitable for publication in PLOS ONE. Congratulations! Your manuscript is now with our production department. 

Kind regards, 

on behalf of

Dr. Anil Kumar Singh 

Academic Editor

PLOS ONE